# ULTRA-FAST INVERSE TONE MAPPING VIA GAIN MAP-BASED LUT

## ABSTRACT

We aim to introduce Look-Up Tables (LUTs), a highly efficient approach, for ultra-fast inverse tone mapping (ITM). However, as LUT size scales exponentially with increasing bit-depth, it remains challenging to employ dense sampling for high bit-depth accuracy. This inevitably introduces quantization artifacts and degrades the fidelity of ITM. To address this issue, we propose GMLUT, which encodes high-bit-depth HDR information into a low-bit-depth learnable Gain Map (GM), thereby facilitating the application of LUTs. Nevertheless, since the LUT alone can only perform global mapping, it is insufficient to address local tone-mapping degradations in practical scenarios. Thus, we devise three closely co-ordinated operators to address the limitation of LUTs: (a) bilateral grids for local adaptation, (b) image-adaptive LUTs for SDR-to-GM translation, and (c) a lightweight neural modulator for GM refinement. In addition, we construct a synthetic dataset of over 8,000 4K SDR–GM pairs together with a real-capture test set to support the training and evaluation of GMLUT. Experiments demonstrate that GMLUT outperforms prior state-of-the-art lightweight ITM methods by +1.4 dB in PQ-PSNR while reducing inference time by 70%. Remarkably, it processes 4K inputs in only 6.2 ms, achieving significant gains in both accuracy and efficiency.

## 1 INTRODUCTION

Modern TVs and devices are increasingly equipped with high dynamic range (HDR) and wide color gamut (WCG) displays, capable of showing brighter and more vivid images. However, most existing content is still in standard dynamic range (SDR), designed for older screens with lower brightness and color range. This gap creates a strong demand for inverse tone mapping (ITM), which converts SDR content into HDR format to enhance visual quality. Unlike professional remastering, practical ITM must run in real-time on high-resolution images, often on edge devices with limited computing power, such as smart TVs or set-top boxes.

Recent learning-based methods (Chen et al., 2021c; Guo et al., 2023a; Chen et al., 2021a; He et al., 2022; Xu et al., 2022; Wang et al., 2022) have demonstrated promising perceptual performance for ITM by directly regressing PQ-compressed or linear HDR values from SDR inputs. These models handle complex tone and color adjustments and perform well on a wide range of content. However, their high computational costs, memory usage, and latency make them unsuitable for real-time inference, particularly on low-power edge devices.

In contrast, look-up tables (LUTs) provide excellent runtime efficiency by replacing computation with indexed queries and interpolation. They are widely used in real-time image enhancement and color grading due to their low latency and deployment simplicity (Adobe Systems Inc., 2005; Blackmagic Design, 2020; Xu et al., 2020). This inherent efficiency makes them appealing for achieving real-time, high-resolution SDR-to-HDR upconversion. However, vanilla LUTs are ill-suited for ITM, particularly in converting low-bit-depth SDR inputs to high-bit-depth HDR outputs. First, achieving sufficient precision for HDR/WCG fidelity requires dense sampling, which scales the table size cubically and leads to increased memory and bandwidth costs. Second, LUTs are context-agnostic, performing pixel-wise transformation from the input $(r, g, b)$ triplet without access to spatial or global information. This severely limits their ability to handle local tone-mapping degradations in SDR content, where adaptive compression is often applied to preserve perceptual detail, especially in highlights and shadows.

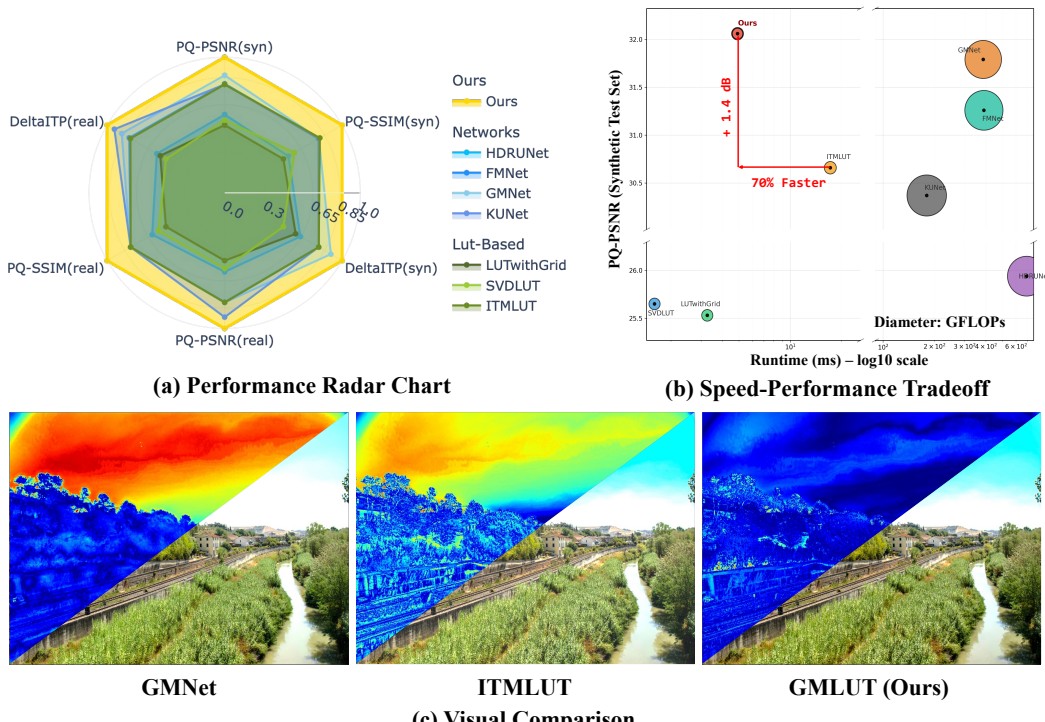

**(a) Performance Radar Chart**

**(b) Speed-Performance Tradeoff**

**(c) Visual Comparison**

Figure 1: (a) Performance radar chart for both synthetic and real test sets. (b) Speed–performance trade-off: GMLUT achieves a 1.4 dB PQ-PSNR gain and a 70% runtime reduction compared with prior lightweight methods. (c) Visual comparisons with baselines, demonstrating superior ITM quality at significantly lower computational cost.

While recent LUT-based methods enhance LUTs capability by learning parameters or mixtures (Yang et al., 2022a; Li et al., 2023; Zeng et al., 2022; Kim et al., 2024; 2025; Li et al., 2022), they remain tailored to 8-bit SDR enhancement, and face key limitations when applied to the more challenging ITM task, including the need for dense sampling and the per-pixel transformation. Notably, ITMLUT (Guo et al., 2023b) extends this direction by predicting three LUTs for dark, midtone, and highlight regions using global network-generated weights. While this improves adaptivity, it still incurs nontrivial computational overhead and remains vulnerable to locally tone-mapped SDR, limiting its effectiveness for real-world ITM deployment.

To achieve real-time ITM in real-world applications, we introduce GMLUT, an ultra-fast framework that leverages a Gain Map-based LUT architecture. Inspired by recent industry standards (ISO; Adobe, 2024; Google, 2024), GMLUT avoids direct regressing high-bit-depth HDR radiance values by predicting an 8-bit color GM, which encodes the HDR and WCG information over the SDR base. This formulation simplifies optimization, mitigates banding artifacts, and aligns with pioneer HDR content formats. To address local tone-mapping degradations while maintaining efficiency, GMLUT generates three image-adaptive operators conditioned on a global thumbnail of the original high-resolution SDR image: (1) a bilateral grid for local adaptation, (2) learnable LUTs for fast SDR-to-GM translation, and (3) a lightweight neural modulator for GM refinement. These operators are sequentially applied to the high-resolution SDR input to produce the GM output, effectively restoring local tone mapping degradation and achieving high-quality ITM with a cache-friendly LUT size. Since high-resolution processing is network-free, the proposed pipeline achieves very low runtime and computational cost.

To facilitate training and comprehensive evaluation, we construct a dataset containing over 8K synthetic SDR–Gain Map pairs based on RAISE raw data (Dang-Nguyen et al., 2015), along with a real-captured test set that reflects practical degradations. As shown in Fig. 1, GMLUT runs at merely 6.20 ms per 4K image on an NVIDIA V100 GPU, achieving a 1.4 dB gain in PQ-PSNR over prior lightweight ITM baselines, while reducing runtime by 70%. We summarize our principal contributions as follows:

- We introduce GMLUT, an ultra-fast GM-based LUT architecture that learns an 8-bit color GM instead of high-bit-depth HDR values, simplifying optimization and mitigating quantization artifacts.

- We efficiently address local tone-mapping degradations by predicting three image-adaptive operators: bilateral grids for local adaptation, learnable LUTs for fast SDR-to-GM translation, and a neural modulator for GM refinement. This introduces spatial adaptivity without increasing LUT resolution or incurring significant computational overhead.

- To support color GM supervision and local tone-mapping degradations, we construct a high-quality dataset of over 8K synthetic SDR–GM pairs along with a real-captured test set for training and comprehensive evaluation.

- Extensive experiments demonstrate that GMLUT achieves a strong quality-efficiency balance, outperforming prior lightweight ITM methods by 1.4 dB in PQ-PSNR while reducing runtime by 70%, requiring only 6.20 ms per 4K image on an NVIDIA V100 GPU.

## 2 RELATED WORK

### 2.1 LEARNING-BASED INVERSE TONE MAPPING

Early learning-based ITM methods typically train neural networks to directly predict PQ-compressed or linear HDR representations from SDR inputs. DeepSR-ITM (Kim et al., 2019), which formulates ITM and super-resolution as a joint learning task to simultaneously enhance resolution and dynamic range; HDRUNet (Chen et al., 2021a) adopts a U-Net architecture tailored for HDR reconstruction, with a focus on structural preservation and highlight details; FMNet (Xu et al., 2022) introduces frequency-aware modulation to suppress low-frequency artifacts and enhance perceptual fidelity; HDRTVNet (Chen et al., 2021c) leverages hierarchical context modeling and multi-scale supervision to improve spatial adaptivity and highlight reconstruction; KUNet (Wang et al., 2022) integrates adaptive kernel selection to capture local variations and improve detail-aware ITM. While these models deliver high perceptual quality, their high FLOPs, memory usage, and latency make them impractical for real-time deployment on consumer-grade edge devices.

### 2.2 GAIN MAP FOR INVERSE TONE MAPPING

Gain Map (GM) is an emerging HDR representation that decouples a high-bit-depth HDR image into a two-layer, low-bit-depth SDR–GM pair, enabling display-adaptive rendering, GPU-friendly processing, and backward compatibility with existing software (ISO; Adobe, 2024; Google, 2024). Building on this format, GMNet (Liao et al.) shifts the prediction target from absolute HDR values to GMs. Compared with direct HDR regression, learning in the GM domain yields a more balanced distribution and preserves highlight details more effectively, making GM a compelling alternative supervision signal for ITM. However, the prior method typically restrict GMs to a single luminance channel, construct datasets accordingly, and rely on heavy neural backbones that are unsuitable for real-time, high-resolution inference. To overcome these limitations, we construct a dataset of over 8,000 high-resolution SDR–color GM pairs to enable color GM supervision. Building on this, our proposed GMLUT predicts a three-channel color Gain Map for both HDR and WCG expansion, integrated with lightweight, image-adaptive operators for efficient and high-fidelity ITM.

### 2.3 LUT-BASED IMAGE PROCESSING

LUTs have long been used for real-time color grading and tone mapping due to their low latency and hardware efficiency. With learning-based extensions, they have been applied to many image processing tasks via basis mixtures (Zeng et al., 2020), dynamic range enhancement with adaptive interval sampling (Yang et al., 2022b), higher-dimensional encoding for contextual variation (Chen et al., 2021b), and spatially adaptive tone mapping guided by feature extractors or bilateral grids (Wang et al., 2020; Gharbi et al., 2017; Kim et al., 2024). To reduce storage, decomposition and compact table have also been explored (Kim et al., 2025; Li et al., 2024). Nonetheless, most LUT-based methods are restricted to 8-bit SDR with low sampling densities, limiting HDR fidelity. High-quality ITM demands denser LUTs or higher-precision coefficients, but their cubic growth strains memory and bandwidth. Moreover, vanilla LUTs are per-pixel and context-agnostic, making them ineffective

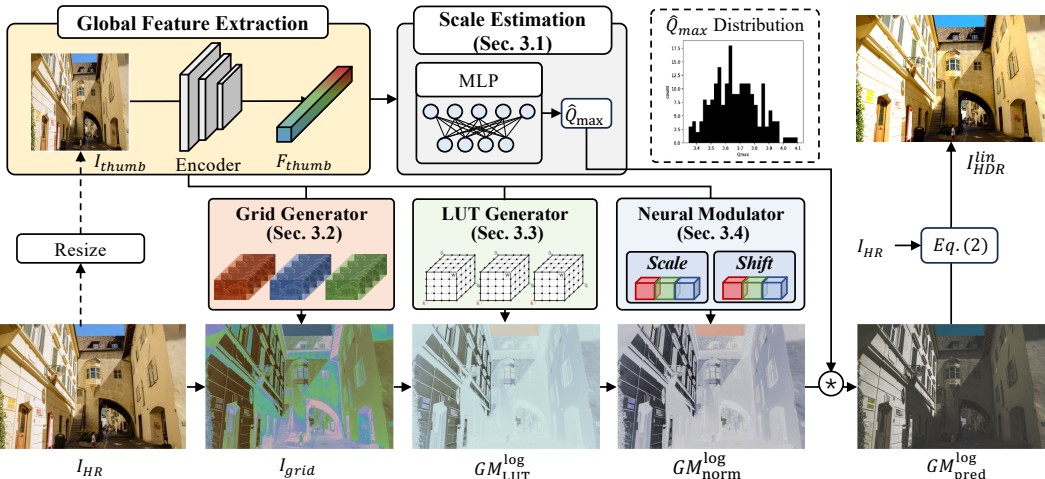

Figure 2: Overview of GMLUT. A high-resolution SDR input $I_{HR}$ is downsampled to a thumbnail $I_{thumb}$, which predicts the global absolute dynamic range $\hat{Q}_{max}$ and generates three image-adaptive operators: a bilateral grid for local adaptation, a 3D LUT for SDR-to-GM translation, and a neural modulator for GM refinement. These operators are sequentially applied to $I_{HR}$ to produce the normalized log-GM $GM_{norm}^{log}$, which is then scaled by $Q_{max}$ to obtain the final GM. The HDR output is reconstructed from the SDR input and the estimated GM (Eq. 2). The figure also illustrates intermediate results and visualizes the predicted $\hat{Q}_{max}$ distribution on the synthetic test set.

for local tone-mapping degradations. Hybrid models such as ITMLUT (Guo et al., 2023b) enhance adaptivity via multiple LUTs from global features, but remain costly and inadequate. Our method retains LUT efficiency while addressing these limitations through globally conditioned, lightweight dynamic operators.

## 3 METHOD

**Overview**    Instead of regressing linear HDR values or compressed counterparts (e.g., tanh, PQ), we predict a *normalized, log-encoded, three-channel* Gain Map $GM_{norm}^{log} \in [-1, 1]$, along with a global scalar $\hat{Q}_{max}$ that defines the absolute dynamic range. The proposed framework generates image-adaptive operators in parallel on the thumbnail version of the high-resolution SDR input, while the generated operators sequentially perform network-free operations on the original SDR to achieve minimal runtime and computational overhead.

Specifically, given a high-resolution SDR image $I_{HR}$, we extract a global feature from the corresponding resized $256\times256$ thumbnail $I_{thumb}$, which is used to predict both $\hat{Q}_{max}$ and three image-adaptive operators: (1) bilateral grids for local adaptation, (2) learnable LUTs for fast SDR-to-GM translation, and (3) a lightweight neural modulator for GM refinement. The dynamic operator are generated in parallel and the sequentially applied to the original high-resolution $I_{HR}$ to generate the normalized log-GM $GM_{norm}^{log}$. The final absolute log-domain GM $GM_{pred}^{log}$ is given by:

$$GM_{pred}^{log} = \hat{Q}_{max} \cdot GM_{norm}^{log}. \tag{1}$$

This formulation explicitly decouples global illuminance prediction ($\hat{Q}_{max}$) from relative dynamic range and color gamut modeling ($GM_{norm}^{log}$). To reconstruct the final HDR output, we follow the standard formulation for log-domain GM. Given the gamma-compressed SDR image $I_{HR} \in [0, 1]$, the predicted log-domain GM $GM_{pred}^{log}$, the final recovered linear HDR image $I_{HDR}^{lin}$ is computed as:

$$I_{HDR}^{log} = \log_2\big((I_{HR})^\gamma + \text{offset}\big) + GM_{pred}^{log}, \quad I_{HDR}^{lin} = \exp(2, I_{HDR}^{log}) - \text{offset}, \tag{2}$$

where the $\gamma$ is set as 2.2, offset is defaulted as 1/64. This fomulation aligns with industry-standard display pipelines (*e.g.*, Adobe and Google (Adobe, 2024; Google, 2024)).

### 3.1 SCALE ESTIMATION

Estimating absolute luminance from cropped patches is inadequate and ill-posed, hindering both learning and accurate recovery. We therefore estimate the global illuminance scale only from the whole image. As shown in Fig. 2, we feed the thumbnail image $I_{\text{thumb}}$ to a small encoder with strided convolutional blocks to produce the global feature $F_{\text{thumb}}$. A two-layer MLP then maps $F_{\text{thumb}}$ to the estimated scale $\hat{Q}_{\max}$. During training, though supervision is on cropped patches, we still provide the full-image thumbnail $I_{\text{thumb}}$ so that the scale remains globally consistent rather than patch dependent and avoid train-test inconsistency.

### 3.2 GRID GENERATION AND SLICING

To address spatially varying local tone mapping degradations, GMLUT predicts dynamic bilateral grids from the thumbnail feature $F_{\text{thumb}}$. While bilateral grids have been widely explored for image enhancement (Gharbi et al., 2017; Kim et al., 2024; 2025), we condition them directly on global features, enabling adaptation to scene-dependent local distortions such as non-uniform tone curves. Specifically, a lightweight MLP generates $K$ grids of dimension $N_b \times N_b \times N_b$; in practice, we use $K = 3$ and $N_b = 8$ to balances expressiveness and efficiency. Using multiple grids mitigates over-smoothing and provides richer, scene-adaptive bases. To avoid extra computation, the input RGB channels are reused as the range guidance. Grid features are then fused with the input RGBs through a $1 \times 1$ projection, producing a spatially modulated base $I_{grid}$ for subsequent LUT transformation.

### 3.3 LUT GENERATION AND TRANSFORMATION

In parallel to the bilateral grid, we generate 3D LUTs from the global thumbnail feature $F_{\text{thumb}}$ to achieve fast SDR-to-GM translation. Specifically, A lightweight two-layer MLP predicts the table parameters $T$ of dimension $3 \times N_t \times N_t \times N_t$, with $N_t$ is set as 17 in our implementation. The generated LUTs directly processes the intermediate high-resolution grid-sliced $I_{grid}$ through tri-linear sampling in $T$, producing a global intensity and color remapping from SDR domain to log-GM domain. Conditioned on global thumbnail features, the generated LUT provides a scene-aware transformation that works jointly with the generated bilateral grid operator, resulting in a refined high-resolution GM $GM_{\text{LUT}}^{\log}$ prior to the final modulation stage.

### 3.4 NEURAL MODULATION

To further incorporate global context into the GM prediction without expensive spatial alignment, we extract compact vectors from the thumbnail feature $F_{\text{thumb}}$ using a small MLP which predicts channel-wise $(\alpha, \beta)$ parameters. The vectors are broadcast over the full resolution and applied to the LUT transformed log-GM $GM_{\text{LUT}}^{\log}$ through an efficient affine transformation:

$$GM_{\text{norm}}^{\log} \;=\; \tanh\bigl(GM_{\text{LUT}}^{\log} \odot (1 + \alpha) + \beta\bigr). \tag{3}$$

This lightweight modulation promotes coherence across large-scale structures (*e.g.*, sky, indoor lighting) while keeping computation low. The final GM prediction is obtained according to Eq. 1.

### 3.5 LOSS FUNCTION

We minimize the objective function with two data-fidelity terms and two LUT regularizers. For Gain Map learning, we use per-pixel $\ell_1$ losses: one on the predicted normalized log domain Gain Map $GM_{\text{norm}}^{\log}$ against its normalized reference $GM_{\text{gt-norm}}^{\log}$ to enforce relative intensity, and another on the final scaled Gain Map $GM_{\text{pred}}^{\log}$ against the original unnormalized reference $GM_{\text{gt}}^{\log}$ to enforcing global scale stably. Following common practice, the LUT is constrained by a smoothness penalty $L_s$ and a monotonicity penalty $L_m$. The overall objective is

$$\mathcal{L} = \|GM_{\text{norm}} - GM_{\text{norm}}^{\text{gt}}\|_1 + \lambda_1 \|GM_{\text{pred}} - GM_{\text{orig}}^{\text{gt}}\|_1 + \lambda_2 (L_s + L_m), \tag{4}$$

where $\lambda_1$ is set as 3 to emphasize both absolute scale estimation and the relative dynamic range prediction, and $\lambda_2$ is set as 0.1 to ensure the LUT monotonicity and smoothness.

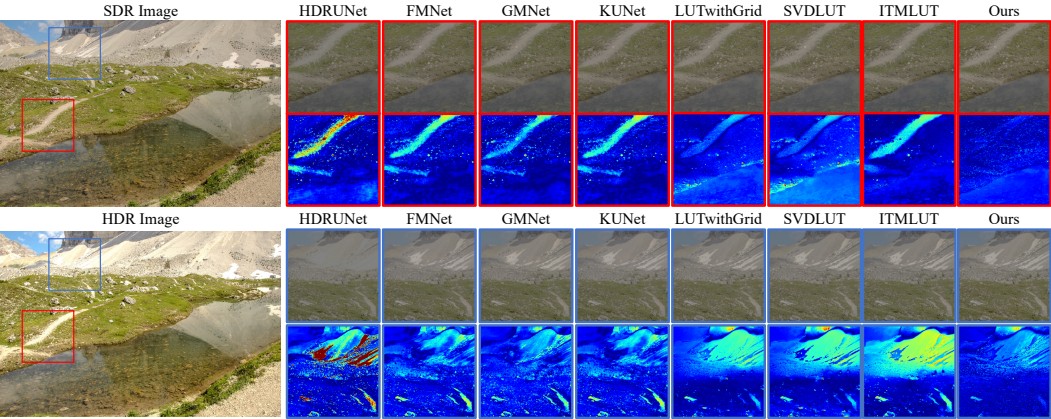

Figure 3: Visual Comparison on Synthetic Test Set. Each column shows the predicted HDR result and its corresponding Y-channel error map (bottom) computed on PQ-encoded HDR. Our method produces more accurate and perceptually faithful results than existing methods.

Table 1: Quantitative comparison on the synthetic dataset (↑ higher is better, ↓ lower is better).

| Methods | Linear domain | | | PQ domain | | | HDR metrics | |
|---|---|---|---|---|---|---|---|---|
| | PSNR↑ | SSIM↑ | SRSIM↑ | PSNR↑ | SSIM↑ | SRSIM↑ | $\Delta E_{\text{ITP}}$ ↓ | HDRVDP3↑ |
| HDRUNet | 34.28 | 0.9426 | **0.9937** | 25.94 | 0.9182 | 0.9836 | 20.08 | 8.691 |
| FMNet | 34.66 | 0.9515 | 0.9899 | 31.26 | 0.9422 | 0.9959 | 15.25 | 9.364 |
| KUNet | 33.95 | 0.9479 | 0.9911 | 30.37 | 0.9422 | **0.9969** | 15.37 | 8.844 |
| GMNet | 33.78 | 0.9439 | 0.9902 | 31.79 | 0.9465 | 0.9966 | 14.08 | **9.385** |
| LUTwithGrid | 34.22 | 0.9407 | 0.9891 | 25.53 | 0.9174 | 0.9803 | 20.64 | 9.069 |
| SVDLUT | 34.23 | 0.9402 | 0.9888 | 25.65 | 0.9203 | 0.9795 | 21.42 | 9.031 |
| ITMLUT | 33.81 | 0.9483 | 0.9888 | 30.66 | 0.9481 | 0.9920 | 15.13 | 9.171 |
| **Ours** | **35.25** | **0.9554** | 0.9912 | **32.06** | **0.9516** | 0.9927 | **13.89** | 9.263 |

## 4 EXPERIMENT

**Proposed Dataset**  Public paired datasets containing SDR images and three-channel (RGB) Gain Maps are unavailable. Existing resources are limited in various ways: HDRTV1K (Chen et al., 2021c) (1235 pairs) and HDRTV4K (3878 pairs) (Guo et al., 2023a) focus on global tone mapping degradation and do not provide per-channel Gain Maps; GMNet (Liao et al.) introduces SDR–Gain Map pairs, but the gain is restricted to a single luminance channel and the scale is modest (900 synthetic and 900 real).

To enable our learning strategy, we construct a new dataset with 8150 4K synthetic SDR–GM pairs for training and testing and 82 real captured pairs for comprehensive evaluation. For the synthetic set, we start from the RAISE RAW image dataset (Dang-Nguyen et al., 2015) and apply adaptive local tone mapping in Adobe Photoshop (Adobe Inc., 2024) to produce SDR renders with their corresponding three-channel Gain Maps. For the real set, we use Adobe Indigo (Levoy & Kainz, 2025), which captures, aligns, and fuses multi-exposure raw images to create HDR content, and directly produces a double-layer SDR–Gain Map format for restoring. All real scenes were recorded with an iPhone 12 Pro Max, spanning daytime/nighttime and indoor/outdoor settings with realistic mobile-photography characteristics. We train on a subset of the synthetic data, reserve 200 synthetic pairs for testing, and use the entire real set only for testing to assess robustness. More details of the proposed datasets can be found in the appendix.

**Implementation Details**  We adopt the Adam optimizer (Kingma & Ba, 2015) with $\beta_1$=0.9 and $\beta_2$=0.99. The learning rate is initialized to $2 \times 10^{-4}$ and decays by a factor of 0.5 at 200k, 400k, 600k, and 800k iterations following a MultiStep schedule, without warm-up. Training is conducted

Figure 4: Visual Comparison on Real-Capture Test Set. Each column shows the predicted HDR result and its Y-channel error map (bottom) computed on PQ-encoded HDR. Our method achieves accurate recovery in challenging nighttime scenes with extreme dynamic range and motion.

Table 2: Quantitative comparison on the real test set ( ↑ higher is better, ↓ lower is better).

| Methods | Linear domain | | | PQ domain | | | HDR metrics | |
|---|---|---|---|---|---|---|---|---|
| | PSNR↑ | SSIM↑ | SRSIM↑ | PSNR↑ | SSIM↑ | SRSIM↑ | $\Delta E_{\text{ITP}}$ ↓ | HDRVDP3↑ |
| HDRUNet | 32.26 | 0.9415 | 0.9581 | 26.38 | 0.8793 | 0.9674 | 32.51 | 7.995 |
| FMNet | 31.63 | **0.9484** | 0.9576 | 29.56 | 0.9274 | 0.9873 | 25.42 | **8.936** |
| GMNet | 31.71 | 0.9449 | 0.9561 | 29.93 | 0.9280 | 0.9851 | 24.73 | 8.898 |
| KUNet | 31.08 | 0.9443 | 0.9546 | 29.60 | 0.9305 | 0.9872 | 24.92 | 8.833 |
| LUTwithGrid | 32.29 | 0.9431 | **0.9613** | 26.11 | 0.8706 | 0.9583 | 32.80 | 7.855 |
| SVDLUT | 32.33 | 0.9432 | 0.9608 | 26.26 | 0.8736 | 0.9597 | 33.04 | 7.994 |
| ITMLUT | **32.61** | 0.9483 | 0.9604 | 29.59 | 0.9229 | 0.9883 | 25.84 | 8.793 |
| **Ours** | 31.28 | 0.9473 | 0.9575 | **30.02** | **0.9404** | **0.9900** | **24.61** | 8.859 |

on $256 \times 256$ random crops with random horizontal flips and rotations. All experiments are performed on NVIDIA V100 GPUs with a batch size of 16.

## 4.1 COMPARISON WITH BASELINE METHODS

**Compared Methods** We compare with three state-of-the-art network-based ITM methods: HDRUNet (Chen et al., 2021a), FMNet (Xu et al., 2022), and KUNet (Wang et al., 2022), which regress linear or compressed HDR targets. We also include two closely related baselines: ITM-LUT (Guo et al., 2023b), which estimates separate LUTs for highlights and shadows followed by fusion, and GMNet (Liao et al.), the first learning-based method that predicts a single-channel Gain Map. To assess efficiency, we adapt two strong LUT-based image-enhancement methods to the ITM setting, namely LUTwithGrid (Kim et al., 2024) and SVDLUT (Kim et al., 2025). All methods are trained and evaluated under the same setting and data splits.

**Evaluation Metrics** We report results in three domains. In the linear domain, because HDR predictions have highly varying peak values, we normalize both prediction and reference by the ground-truth peak to $[0, 1]$ and then compute PSNR, SSIM, and SRSIM (Zhang & Li, 2012) for comparison. In the PQ domain, we encode prediction and reference with the PQ function and report the same three metrics, which align better with perceived contrast. In the HDR domain, we additionally report the color-difference metric $\Delta E_{\text{ITP}}$ (ITU-R, 2019) and the perceptual quality index HDRVDP3 (Mantiuk et al., 2023).

**Quantitative Evaluation** We evaluate GMLUT on a synthetic test set with 200 pairs and a real-captured HDR test set across linear, PQ, and perceptual HDR domains (Tabs. 1 and 2). On the synthetic dataset, GMLUT achieves new state-of-the-art performance among lightweight ITM methods, surpassing baselines by a significant margin. Notably, it delivers a 1.4 dB improvement in PQ-PSNR

Table 3: Efficiency and quality trade-off analysis on both 4K and 2K resolutions.

| Methods | Params (M) | Runtime (ms, 4K/2K) | FLOPs (G, 4K/2K) | Memory (MB, 4K/2K) | PQ-PSNR (synthetic) | PQ-PSNR (real) |
|---|---|---|---|---|---|---|
| HDRUNet | 1.65 | 785 / 200 | 2946 / 736 | 16435 / 4112 | 25.94 | 26.38 |
| FMNet | 1.30 | 478 / 120 | 2935 / 734 | 5267 / 1326 | 31.26 | 29.55 |
| KUNet | 1.14 | 941 / 237 | 2666 / 667 | 15105 / 3780 | 30.37 | 29.93 |
| GMNet | 1.92 | 455 / 115 | 3155 / 790 | 4955 / 1245 | 31.79 | 29.60 |
| LUTwithGrid | 0.46 | 4.19 / 1.41 | 0.25 / 0.08 | 674.9 / 280.6 | 25.53 | 26.11 |
| SVDLUT | 0.27 | 1.78 / 1.56 | 0.02 / 0.02 | 388.9 / 104.2 | 25.65 | 26.26 |
| ITMLUT | 0.60 | 18.5 / 6.29 | 41.9 / 10.5 | 676.5 / 198.8 | 30.66 | 29.59 |
| GMLUT (Ours) | 0.64 | 6.20 / 2.42 | 0.48 / 0.25 | 962.1 / 330.4 | 32.06 | 30.01 |
| GMLUT (downsample 2048) | 0.64 | 3.07 / 2.45 | 0.26 / 0.25 | 588.3 / 330.4 | 32.21 | 29.98 |
| GMLUT (downsample 1024) | 0.64 | 2.27 / 2.20 | 0.20 / 0.20 | 507.4 / 252.2 | 32.12 | 29.57 |

over ITMLUT. Compared with strong network baselines such as GMNet, our method still attains a +0.27 dB improvement in PQ-PSNR and achieves the best $\Delta E_{ITP}$ score, demonstrating that LUT-based efficiency does not come at the cost of accuracy. The same trend holds on the real set, where GMLUT delivers the highest PQ-PSNR and lowest $\Delta E_{\text{ITP}}$. The advantage becomes more pronounced when compared against LUT-based methods, confirming that conditioning on global features and predicting GMs provides a significant step forward for LUT-based ITM. Overall, GMLUT combines network-level accuracy with LUT-like efficiency. It surpasses state-of-the-art learning-based networks while retaining the lightweight efficiency of LUT-based methods.

**Qualitative Comparison**  As shown in Figs. 3 and 4, visual comparisons further highlight the benefits of GMLUT. Our method preserves fine details in both highlights and shadows, reduces halo artifacts, and maintains natural color reproduction. In contrast, LUT-only methods often produce oversmoothed structures and hue shifts, while network baselines tend to lose sharp local contrast. The combination of a dynamic bilateral grid and image-wise LUT enables sharper local modulation together with faithful color recovery.

**Efficiency Analysis**  We analyze runtime, FLOPs, and memory usage at 4K and 2K resolutions in Tab. 3. At 4K resolution, GMLUT runs in 6.20 ms with a tiny computational footprint, more than 70% faster than ITMLUT and two orders of magnitude faster than network-based baselines. At 2K resolution, latency decreases further to 2.4 ms with proportionally lower FLOPs and memory. In terms of quality, GMLUT consistently outperforms LUT-based methods and slightly surpasses strong network baselines, establishing it as the first approach to combine network-level accuracy with LUT-level efficiency. Further, a practical trade-off is also available: downsampling the input 4k SDR to 2K reduces latency by 51% with negligible loss in quality, while downsampling input 2k SDR to 1K maintains acceptable performance with further savings in runtime and memory.

## 4.2 ABLATION STUDIES

We validate the contribution of each component by removing one module at a time while keeping data, training, and evaluation protocols fixed. As shown in Tab. 4, the ablation results are reported on the real test set using PQ domain metrics and HDR metrics.

**Ablation of Gain Map Learning**  We eliminate the Gain Map learning strategy, including both Gain Map supervision and scale estimation, which reduces the model to direct HDR radiance regression. This severely exacerbates bit-depth limitations, weakens dynamic range disentanglement, increases $\Delta E_{\text{ITP}}$ from 24.61 to 31.89, and lowers PQ-PSNR by more than 3 dB, underscoring the necessity of learning an explicit log-Gain Map. Moreover, the visual comparison in Fig. 5 shows that direct HDR regression makes learning extremely difficult for the LUT, preventing accurate up-conversion in highlight and shadow regions.

**Ablation of Grid Slicing**  In this experiment, we remove grid generation and slicing from our framework. Without the dynamic bilateral grid, the model lacks spatially adaptive bases and relies

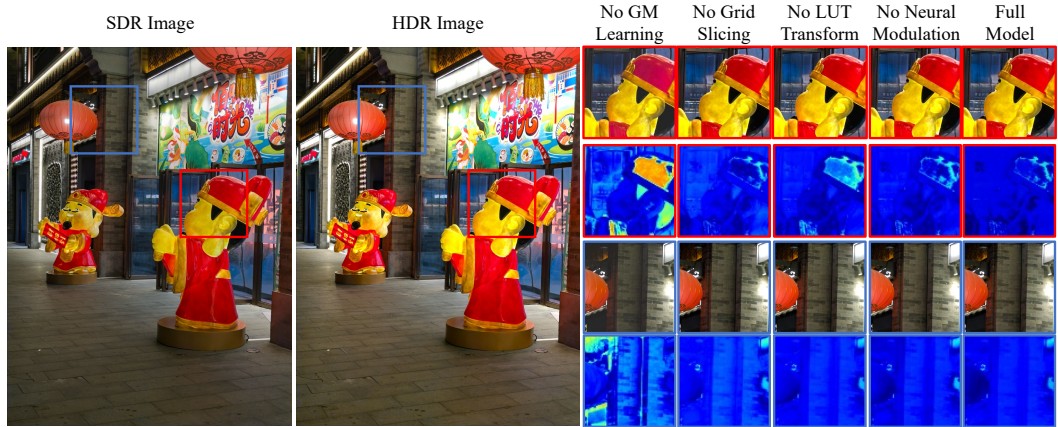

Figure 5: Visual ablation of key components on the real test set.

Table 4: Ablation on major components on real test set (↑ higher is better; ↓ lower is better).

| Configuration | PQ-PSNR↑ | PQ-SSIM↑ | $\Delta E_{ITP}$ ↓ | HDRVDP3↑ |
|---|---|---|---|---|
| w/o Gain Map learning | 26.59 | 0.8823 | 31.89 | 7.934 |
| w/o Grid slicing | 29.68 | 0.9382 | 25.35 | 8.788 |
| w/o LUT transform | 29.75 | 0.9373 | 25.14 | 8.839 |
| w/o Neural modulation | 29.79 | 0.9372 | 25.33 | 8.825 |
| **Full model** | **30.01** | **0.9404** | **24.61** | **8.859** |

solely on the LUT and global modulator. This prevents the framework from handling spatially variant tone-mapping degradations, leading to flattened contrast and weakened edge structures, as reflected by a notable drop across metrics. Visual results in Fig. 5 further confirm this effect.

**Ablation of LUT Transform**  Removing the LUT severely impairs color fidelity. Although the grid preserves spatial structure, both $\Delta E_{ITP}$ and PQ-PSNR drop noticeably, demonstrating that the LUT is essential for channel-wise enhancement and complements spatial modulation.

**Ablation of Neural Modulation**  Disabling the lightweight neural modulation results in minor quality degradation across all metrics. Although the overall performance remains reasonable, the outputs exhibit reduced global consistency. This verifies the modulator's role in enhancing global coherence at negligible computational cost.

**Resolution–Performance Trade-off**  We further evaluate GMLUT under different input resolutions (Tab. 3). At native 4K resolution, the model achieves a runtime of 6.20 ms and a PQ-PSNR of 30.01 dB on the real test set. Reducing the 4K input SDR to 2K resolution decreases runtime to 3.07 ms, corresponding to a 51% speedup, with negligible accuracy loss. Further downsampling 4K input to 1K resolution reduces runtime to 2.27 ms but introduces moderate degradation. These results are relevant for practical deployment: since the Gain Map information is not highly sensitive to resolution, HDR images can be reconstructed from downsampled Gain Maps. Consequently, GMLUT maintains strong perceptual performance while supporting flexible efficiency–accuracy trade-offs.

## 5  CONCLUSION

We propose GMLUT, an ultra-fast Gain Map-based LUT framework for real-time inverse tone mapping. By predicting an 8-bit color GM instead of high-bit-depth HDR values, GMLUT simplifies optimization and alleviates quantization artifacts. To handle local tone-mapping degradations without increasing LUT resolution, it generates three lightweight, image-adaptive operators: a bilateral grid for local adaptation, learnable LUTs for SDR-to-GM translation, and a neural modulator for GM refinement. We further construct a high-quality dataset of over 8K synthetic SDR–GM pairs with a real-captured test set to support color GM supervision and evaluation. Extensive experiments demonstrate that GMLUT achieves superior quality–efficiency trade-offs, surpassing prior lightweight ITM baselines while running in only 6.2 ms for 4K inputs.

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

# A APPENDIX

## A.1 MORE VISUAL COMPARISON

We further provide more visual comparison on both the synthetic dataset and the real-capture test set in Fig. 6 and 7, respectively. It is clear that GMLUT is able to produce more accurate illuminance recovery and gamut expansion under diverse conditions for practical usage.

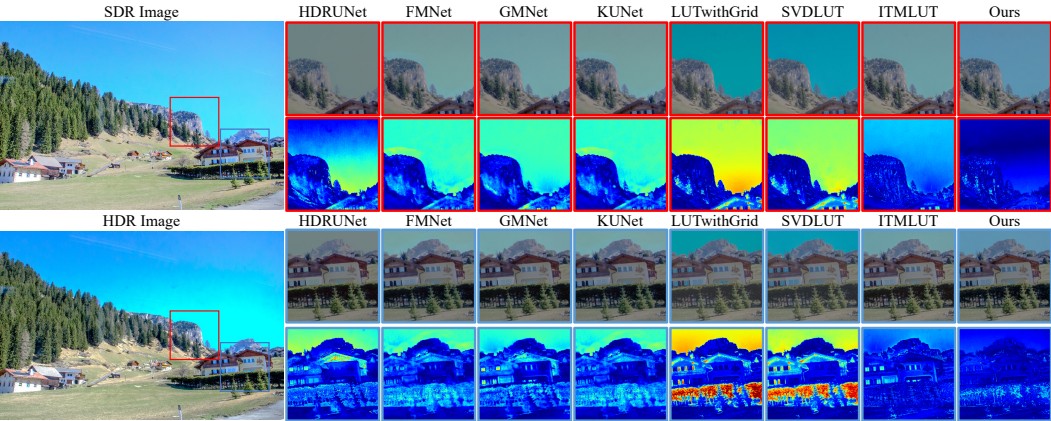

Figure 6: Visual Comparison on Synthetic Test Set. Each column shows the predicted HDR result and its corresponding Y-channel error map (bottom) computed on PQ-encoded HDR. Our method produces more accurate and perceptually faithful results than existing methods.

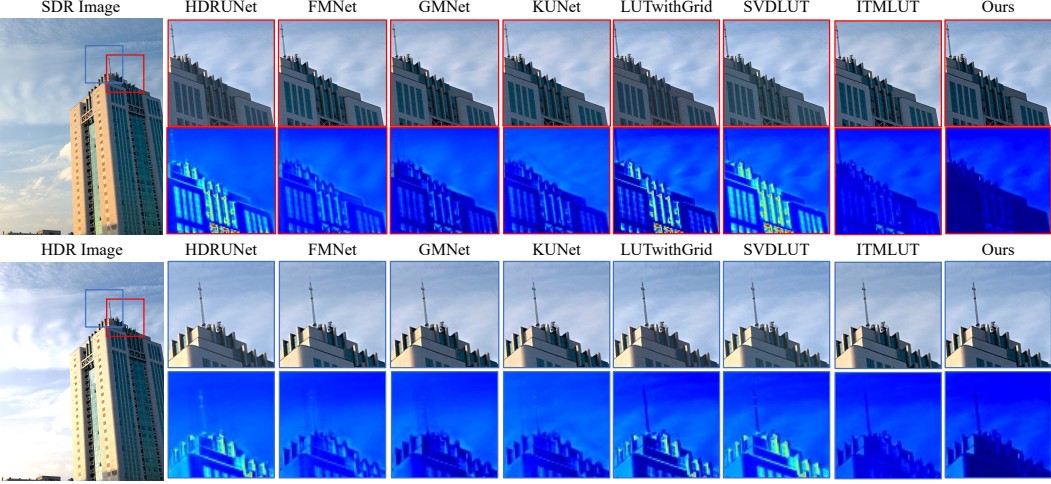

Figure 7: Visual Comparison on Real Capture Test Set. Each column shows the predicted HDR result and its Y-channel error map (bottom) computed on PQ-encoded HDR.

## A.2 INTERMEDIATE STAGES VISUALIZATION

We provide additional visualizations of intermediate outputs from different stages of GMLUT to illustrate its working process in Fig. 8. The generated bilateral grid first performs fast local adaptation on the input SDR image, while remaining in the linear SDR domain. The learnable LUT then translates the adapted SDR to an initial GM. Next, the neural modulator corrects the global scale and shift for each channel to ensure accurate estimation. Finally, the normalized GM is combined with the estimated absolute dynamic range to produce the final GM. Since none of these operators involve high-resolution network processing, the runtime and computational overhead remain minimal.

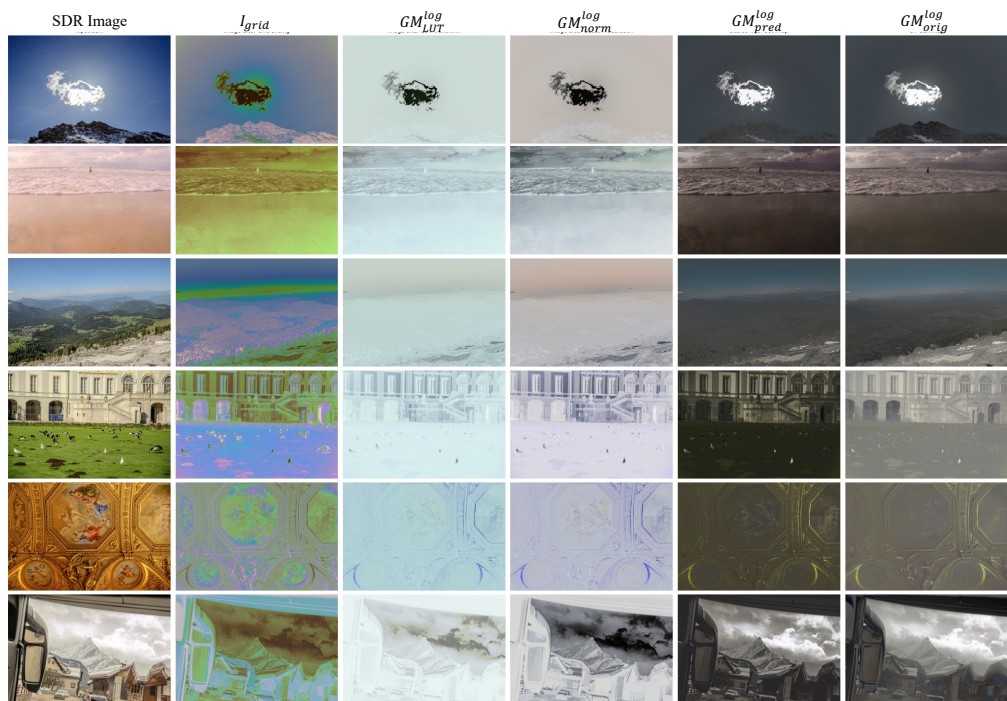

Figure 8: Intermediate results of GMLUT. We illustrate the intermediate stages of the GMLUT process, including bilateral grid processing, LUT translation, and neural modulation. For clear visualization, we also present the final predicted log-domain GM alongside the corresponding ground truth log-domain GM.

## A.3 DATASET DETAILS

We visualize representative samples from the constructed synthetic dataset and the real-capture mobile dataset in Fig. 9 and 10, respectively. The synthetic dataset is derived from the RAISE dataset Dang-Nguyen et al. (2015), which contains 8,156 high-resolution, uncompressed RAW images collected over three years by four photographers using three different cameras across more than 80 European locations. All images are guaranteed to be camera-native and unaltered. We process the RAW data in Adobe Photoshop to generate SDR–GM pairs by applying the "HDR" function with "Auto" settings and limiting the "HDR" parameter to three, thereby producing locally tone-mapped SDR base images. The SDR–GM pairs are saved in JPEG format with the P3 color gamut.

The real-captured dataset is acquired using an iPhone 12 Pro Max with Adobe Indigo, an experimental computational photography system. Indigo merges up to 32 underexposed RAW frames to preserve highlights and suppress noise, followed by mild local tone mapping with semantically aware adjustments, resulting in natural-looking images. It outputs paired SDR base and GainMap images in JPEG format. The real-captured test set contains 82 SDR–GM pairs covering diverse conditions (day/night, indoor/outdoor, and dynamic scenes with slight blur). Reference HDR images can be reconstructed from the paired SDR and GM for quantitative evaluation.

## A.4 AI USE DECLARATION

We used Qwen-3 solely for grammar checking in Sections 3 and 4. No large language model was employed for generating experimental results, figures, or core technical claims. All content was reviewed, verified, and approved by the human authors.

## A.5 ETHICS STATEMENT

We confirm that this work adheres to the ICLR Code of Ethics. It does not involve human subjects, sensitive data, or practices that raise ethical concerns.

## A.6 REPRODUCIBILITY STATEMENT

We detail the components of GMLUT in Sec. 3 and the experimental settings in Sec. 4, and provide extensive visual results in Sec. 4 and the appendix for verification. We will release the training and testing code together with the constructed datasets upon publication to ensure full reproducibility.

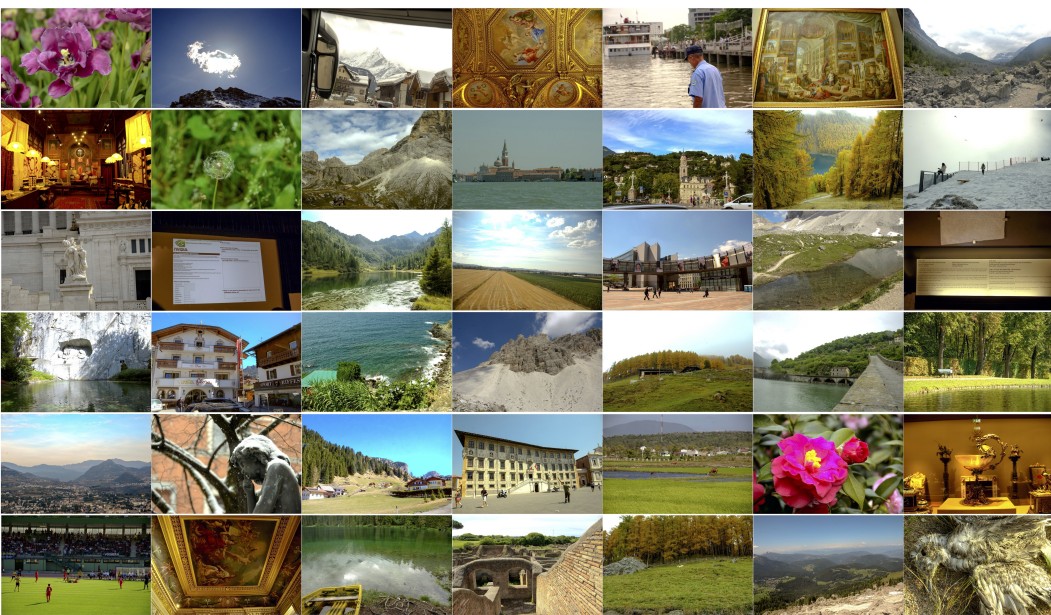

Figure 9: Visualization of the synthetic dataset. The synthetic dataset contains 8,000+ 4K SDR–GM pairs covering landscapes, nature, people, objects, and assorted indoor/outdoor scenes, delivering high-quality training data for practical ITM methods.

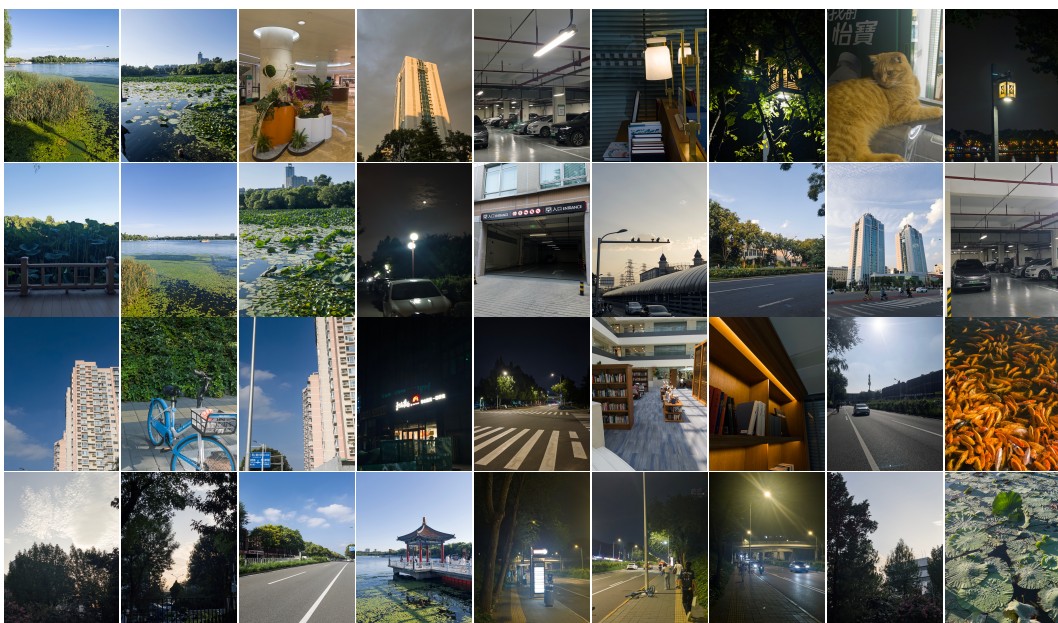

Figure 10: Visualization of the real-capture mobile dataset by iPhone 12 Pro Max with Adobe indigo software. The dataset spans diverse conditions, including nighttime and daytime, indoor and outdoor scenes, as well as static and dynamic content.

