# OpenReview forum: "Ultra-Fast Inverse Tone Mapping via Gain Map-based LUT"
_ICLR.cc/2026/Conference — ICLR 2026 Conference Withdrawn Submission_

### Official Review · Reviewer_zMeb · 2025-10-19

**Soundness:** 3
**Presentation:** 3
**Contribution:** 2
**Rating:** 4
**Confidence:** 4

**Summary:**

This paper proposes GMLUT, which learns to transform the standard dynamic range (SDR) images to Gain Maps (GMs) for efficient inverse tone mapping (ITM). This method combines Look-Up Tables (LUTs) for SDR-to-GM transformation, bilateral grids for local adaptation, and a light-weight neural modulator for GM refinement. A notable contribution is the curation of a new dataset consisting of over 8,000 synthetic SDR-GM pairs and a small-scale real-captured test set. The method is demonstrated to be fast and effective on its proposed test sets.

**Strengths:**

**Clarity and Reproducibility:**
The paper is generally written well and easy to follow. The commitment to release both code and dataset strengthens its practical value and reproducibility.

**Dataset Contribution:**
Constructing a large-scale dataset for the challenging ITM task is appreciated, which may facilitate future research in this field.

**Performance-Efficiency Balance:**
The results demonstrate an impressive balance, providing competitive HDR reconstruction quality while maintaining high inference speed for high-resolution images.

**Weaknesses:**

Unfortunately, despite these strengths, the paper currently lacks a strong establishment of its core novelty and rigorous evaluation.

**Weak Motivation and Unclear Technical Novelty:**
The paper builds upon the SDR-to-GM formulation proposed by Liao et al. (2025). The primary extension appears to be the move from a single-channel to a three-channel GM representation. However, the motivation for this design choice is not explained.

***(1)*** Why is a three-channel GM representation necessary or superior? What specific limitations of the single-channel formulation does it address? A clear narrative is missing, making the core contribution feel more like an incremental engineering adjustment than a principled scientific advancement.

***(2)*** The overall architecture is a combination of well-established components (LUTs, bilateral grids, a small neural network). The paper does not articulate a novel insight that justifies this specific assembly. What is the unique conceptual synergy between these elements that solves a key problem in ITM that previous methods could not? Without this insight, the method risks being perceived as a straightforward pipeline of existing techniques rather than a novel solution to this well-established problem.

**Insufficient Evaluation:**
The evaluation is currently confined to the proposed new dataset, which limits the claims of generalizability and state-of-the-art performance.

***(1) Lack of Comparison on Established Benchmarks:*** A critical issue is the failure to evaluate on existing public ITM benchmarks such as HDRTV1K, HDRTV4K, and the dataset of Liao et al. (2025). To make a convincing claim about overall performance and generalization, the method is expected to be tested on these independent datasets, while the performance on the proposed dataset alone is not sufficient to establish the method's broader effectiveness.

***(2) Incomplete and Unconvincing Ablation Study:*** The ablation study in Table 4 is insufficient to reflect the contribution of each component.

**--->** It lacks an important baseline of a single-channel GM variant to justify the three-channel design.

**--->** Simply removing bilateral grids and neural modulator does not justify their novelties. More ablations are required, e.g., comparisons with other/existing bilateral processing mechanisms and internally ablated variants of the neural modulator (e.g., by removing the channel-wise parameters), to highlight the contributions of novel designs. In addition, the Q_max is not involved in the ablations. Is it important?

**--->** Efficiency metrics are missing from the ablation table (Table 4). For a method that highlights efficiency, it is essential to show how each component affects the performance-speed trade-off.


**Dataset Rigor and Broader Impact:**
The new dataset is a strength, but its construction and potential impact require further justification.


***(1) Train-Test Split Justification:***
The extremely high train-test split ratio (around 28:1) is not a common practice and requires explicit justification. A discussion on how the 200 synthetic and 82 real SDR-GM pairs were selected to ensure they are representative of the dataset's diversity and complexity is necessary for a reliable evaluation.

***(2) Broader Utility of the Dataset:***
The dataset seems tailored to the proposed 3-channel GM formulation. To demonstrate its value as a community resource beyond this specific paper, it would be highly impactful to show that this dataset can be used to improve other existing ITM methods. Is it possible to take advantage of this dataset to produce a better performance for other methods?

**Justification for Recommendation:**
This paper presents a practical and efficient method for ITM and provides a valuable new dataset. The performance-efficiency trade-off is compelling.

However, the paper currently lacks a clear narrative regarding its core conceptual novelty and a rigorous enough evaluation to support its claims. The authors are suggested to reframe their contribution around a clearer scientific insight, provide comprehensive evaluations on established benchmarks, and conduct thorough ablations that include efficiency metrics.

**Questions:**

**Conceptual Motivation:** What is the specific hypothesis behind using a three-channel GM representation instead of the single-channel one from Liao et al. (2025)? What quantitative or qualitative advantages does this design offer, and can the authors demonstrate this with a direct comparison?

**Generalization Performance:** To substantiate the claim of state-of-the-art performance, can the authors provide evaluation results on the established benchmarks? This is critical for assessing the method's true capability for HDR reconstruction.

**Ablation Rigor:** Could the authors expand Table 4 to include: (1) a single-channel GM baseline, (2) variants that alter the bilateral grid and neural modulator, (3) a baseline without Q_max, and (4) the corresponding efficiency metrics (e.g., inference time/FLOPs) for each ablated variant? This is essential to ground the design choices in empirical evidence.

**Dataset Impact:** Beyond its use in this paper, have the authors explored using this new dataset for other ITM methods? Is it possible to combine this synthetic dataset with existing real-world data to further boost the performance?

**Details Of Ethics Concerns:**

There is a contradiction between the Ethics Statement (Section A.5, Appendix), which claims "no human subjects are involved," and the caption of Figure 9 (Appendix), which states that people are present in the dataset.

---

### Official Review · Reviewer_PRnR · 2025-10-30

**Soundness:** 3
**Presentation:** 3
**Contribution:** 3
**Rating:** 8
**Confidence:** 5

**Summary:**

This paper proposes GMLUT that predicts Gain Map for high-resolution inverse tone mapping. It employs three image-adaptive operators: bilateral grids, LUT, and neural modulator to address local tone-mapping degradations. Besides, this paper constructs a 8K dataset with SDE-GM pairs for training and evaluation. Extensive experiments demonstrate the effectiveness and efficiency of the proposed GMLUT.

**Strengths:**

1. Instead of HDR values or LUT, the proposed method learns a color gain map, mitigating quantization artifacts. The motivation is interesting, and the experiment results demonstrate this manner can effectively generate high-quality outcomes while requiring minimal computational overhead.
2. This paper constructs a large-scale 8K dataset with SDR-GM pairs, promoting the ITM task.

**Weaknesses:**

1. The proposed GMLUT employs: (a) bilateral grids for local adaptation, (b) image-adaptive LUTs for SDR-to-GM translation, and (c) a lightweight neural modulator for GM refinement. The output of LUT and neural modulator is under supervision, yet the output of bilateral grids misses supervision. How to ensure it meets the expected goals?
2. As the paper noted, LUT suffers from the quantization issue. However, the GMLUT also employs a LUT. How does it address this issue?
3. What are the predicting resolution of the three operators (bilateral grid, LUTs, neural modulator) ? If predicting in a low resolution, how to restore the details? And, a high resolution requires additional computational overhead.

**Questions:**

See weakness.

---

### Official Review · Reviewer_LNr5 · 2025-11-01

**Soundness:** 3
**Presentation:** 3
**Contribution:** 3
**Rating:** 4
**Confidence:** 3

**Summary:**

This paper proposes GMLUT, a gain-map-based LUT framework for real-time inverse tone mapping (ITM). It combines a bilateral grid, an image-adaptive LUT, and a lightweight neural modulator to achieve locally adaptive, ultra-fast HDR reconstruction from SDR inputs. The method reports strong runtime efficiency (6.2 ms on 4 K) and modest performance gains over prior LUT-based and lightweight deep approaches.

**Strengths:**

1. Practical significance: The proposed pipeline achieves exceptional inference speed with low memory and FLOPs, making it highly suitable for deployment on low-power edge devices.

2. Comprehensive experiments: The paper includes extensive quantitative and qualitative comparisons, ablations, and both synthetic and real-capture datasets.

3. Reproducibility and clarity: The technical presentation and dataset description are clear, and the results are consistent across evaluation domains (linear, PQ, HDR metrics).

4. New dataset. This paper introduces a new dataset for ITM

**Weaknesses:**

1. Limited novelty. The method largely integrates existing ideas—Gain-Map representation, bilateral grids, and LUT-based enhancement—into a unified framework. The contribution is mainly engineering-oriented, showing solid system design but few new conceptual insights.

2. Marginal improvement on some datasets. Gains over strong baselines such as GMNet or ITMLUT are sometimes modest (≈ 0.2–0.3 dB in PQ-PSNR) and occasionally lower on real-world scenes. This raises doubt about the generality of the improvement.

3. Missing broader justification. While speed is highlighted, the paper could better discuss trade-offs between model complexity and perceptual quality, or compare with hardware-accelerated real-time HDR solutions.

**Questions:**

Please see Weakness and answer all the concerns in it.

---

### Official Review · Reviewer_W9wc · 2025-11-01

**Soundness:** 2
**Presentation:** 3
**Contribution:** 2
**Rating:** 4
**Confidence:** 5

**Summary:**

This paper introduces GMLUT, a fast and lightweight framework for inverse tone mapping that uses Gain Map encoding together with learnable LUTs to convert SDR images into HDR. Instead of predicting HDR values directly, it estimates an 8-bit Gain Map and applies a few adaptive operators derived from a low-resolution version of the image to restore HDR details efficiently.

**Strengths:**

The paper presents GMLUT, an efficient inverse tone mapping framework that integrates Gain Map encoding with learnable LUTs for real-time SDR-to-HDR conversion. The approach is simple yet effective, delivering high perceptual quality with extremely low computational cost.

**Weaknesses:**

Please refer the Questions.

**Questions:**

If the authors can address the following points, it would greatly help me better understand and evaluate the paper.
1. The overall design appears quite similar to deep bilateral filtering pipelines, where a lightweight network processes a low-resolution thumbnail before propagating results to high resolution. The novelty over existing bilateral grid methods is not clearly explained.
2.It is also unclear which color space (e.g., BT.2020) the constructed HDR dataset uses; a color gamut comparison like Fig. 1 in [GamutMLP CVPR 2023]  would make the work more complete.
3.The authors should clarify whether a single RAW exposure from the RAISE dataset provides sufficient dynamic range for HDR–SDR pair generation, as multi-exposure fusion (e.g., Adobe Indigo) is typically required to capture the full luminance range.
4.Moreover, SDR images often contain dark noise and highlight saturation, as noted in HDRCNN and UltraFusion, but the paper does not analyze how such degradations may affect the bilateral grid’s robustness. Testing on those datasets and adding visualizations could strengthen the analysis.
4.Since the paper claims that Gain Map learning reduces banding artifacts compared with direct HDR regression, an explicit visual comparison would be helpful.
5.Perceptual error visualization could also be improved using HDR-VDP distortion maps instead of raw RGB residuals.
6.Finally, perceptual metrics such as PU-PSNR and PU-SSIM would be more appropriate than standard PSNR/SSIM.
7.Several important references are also missing:

HDR image reconstruction from a single exposure using deep CNNs
UltraFusion: Ultra High Dynamic Imaging using Exposure Fusion
Gain-MLP: Improving HDR Gain Map Encoding via a Lightweight MLP
GlowGAN: Unsupervised Learning of HDR Images from LDR Images in the Wild
LEDiff: Latent Exposure Diffusion for HDR Generation
Revisiting the Stack-Based Inverse Tone Mapping
Single-Image HDR Reconstruction by Learning to Reverse the Camera Pipeline (also built its dataset from RAISE)

---

### Official Review · Reviewer_vEzm · 2025-11-02

**Soundness:** 2
**Presentation:** 2
**Contribution:** 2
**Rating:** 2
**Confidence:** 4

**Summary:**

The paper presents a novel and fast solution for inverse tone mapping (ITM) based on gain map-driven Look-Up Table (LUT). This approach simultaneously outperforms previous state-of-the-art methods in image quality (PQ-PSNR) and offers a substantially faster inference speed.

**Strengths:**

The proposed architecture mitigate the limitations of standard LUTs by proposing a sophisticated approach by including bilateral grids (enabling crucial local adaptation), image-adaptive LUTs, and a neural modulator to improve the inverse tone mapping process. This design choice is empirically validated: the experimental results consistently demonstrate that the proposed method is superior to prior works, not only in terms of objective image quality but also in its remarkable efficiency and low computational complexity.

**Weaknesses:**

1. The entire pipeline relies heavily on features extracted from a thumbnail image which are subsequently used across multiple specialized modules (grid generation, LUT generation, and neural modulation). However, the specific design and structure of the encoder responsible for generating these features are entirely missing. Given that each subsequent module has a distinct functional purpose (e.g., generating spatial grids vs. generating color mapping parameters), they likely require specialized or differentiated feature representations. The absence of details on the encoder design makes it impossible to evaluate whether the extracted features are suitable for these diverse tasks.
2. The mathematical expression in the second part of Equation (2) needs to be clarified as currently presented, it is difficult to interpret the intended operation.
3. This paper does not clearly explain the interdependence between the grid generation and LUT generation modules. A detailed mathematical or logical description of how the outputs of one module inform the inputs or constraints of the other is necessary. More details, including the specific mathematical operations, are required for the grid generation module.
4. The experiments are limited to a single dataset that was constructed internally by the authors. For a convincing demonstration of a state-of-the-art inverse tone mapping technique, validation against at least one widely recognized, publicly available benchmark dataset is essential.

**Questions:**

1.Ablation Network Architecture: The corresponding network architectures (pipeline diagrams) for the specific ablation experiments are missing. For example, when evaluating the impact of the grid/LUT components, how exactly was the rest of the architecture modified or removed?
2. Thumbnail Image Size: The overall complexity is tied to the use of a thumbnail image for feature extraction. Please provide an analysis and discussion on the impact of varying the input size of the thumbnail image used by the encoder. Did the authors explore different sizes, and how did this affect the trade-off between speed and PQ-PSNR?
3. LUT Size: Similarly, the size of the Look-Up Table (LUT) is a core parameter. Please discuss the motivation behind the chosen LUT size and provide an ablation on how different LUT dimensions influence both the mapping precision (quality) and the resulting inference time (complexity).

---

### Note · Authors · 2025-11-14

**Comment:**

Thanks for all valueable comments and advice, we will further clarify and improve our work.

**Withdrawal Confirmation:**

I have read and agree with the venue's withdrawal policy on behalf of myself and my co-authors.